# Metformin Triggers Apoptosis and Induction of the G0/G1 Switch 2 Gene in Macrophages

**DOI:** 10.3390/genes12091437

**Published:** 2021-09-17

**Authors:** Xuming Hu, Huan Luo, Chunfeng Dou, Xujing Chen, Yi Huang, Liping Wang, Songlei Xue, Zhen Sun, Shihao Chen, Qi Xu, Tuoyu Geng, Xin Zhao, Hengmi Cui

**Affiliations:** 1Institute of Epigenetics and Epigenomics, College of Animal Science and Technology, Yangzhou University, Yangzhou 225009, China; hxm@yzu.edu.cn (X.H.); LH13952752374@163.com (H.L.); calthusdcf@gmail.com (C.D.); chenxujing1126@163.com (X.C.); NJPHWLP@163.com (L.W.); xuesl2013@hotmail.com (S.X.); frostmournk@163.com (Z.S.); mrrchen@yzu.edu.cn (S.C.); tygeng@yzu.edu.cn (T.G.); 2Department of Animal Science, McGill University, Montréal, QC H3A 0G4, Canada; xuqi@yzu.edu.cn; 3Department of Pharmacy, Suzhou Vocational Health College, Suzhou 215009, China; huangyicpu@126.com; 4Joint International Research Laboratory of Agricultural & Agri-Product Safety, Ministry of Education of China, Yangzhou University, Yangzhou 225009, China; 5Jiangsu Co-Innovation Center for Prevention & Control of Important Animal Infectious Diseases & Zoonoses, Yangzhou 225009, China

**Keywords:** macrophages, metformin, apoptosis, G0S2, Bcl-2

## Abstract

Metformin is a widely used antidiabetic drug for the treatment of type 2 diabetes and has been recently demonstrated to possess anti-inflammatory properties via AMPK-mediated modulation of M2 macrophage activation. However, the anti-inflammatory mechanisms of metformin on inflammatory macrophages are still not fully elucidated. In this study, we found that metformin induced apoptosis in macrophages. In particular, metformin induced apoptosis of M1 macrophages, based on M1 marker genes in apoptotic macrophages. Next, we comprehensively screened metformin-responsive genes in macrophages by RNA-seq and focused on the extrinsic apoptotic signaling pathway. The G0/G1 switch 2 gene (G0S2) was robustly up-regulated by metformin in macrophages. Overexpression of G0S2 significantly induced apoptosis of macrophages in a dose-dependent manner and blunted the function of the crucial anti-apoptotic gene Bcl-2, which was significantly reduced by metformin. These findings show that metformin promoted apoptosis of macrophages, especially M1 macrophages, via G0S2 induction and provides a novel anti-inflammatory mechanism of metformin through induction of macrophage apoptosis.

## 1. Introduction

Macrophages have a defensive function against pathogens such as microbes and play an important role in homeostatic maintenance of the body through disposal of internal waste materials and tissue repair [1,2]. To perform these functions, macrophages modify their own metabolism and phenotypes. Phenotypically polarized macrophages are now generally recognized as pro-inflammatory “classically” activated M1 macrophages and “alternatively” activated M2 macrophages [3]. M1 macrophages exhibit a pro-inflammatory response, with high production of pro-inflammatory cytokines (IL-1β, IL-6 and TNF-α), and are implicated in initiating and sustaining inflammation. On the other hand, M2 macrophages possess anti-inflammatory properties and are involved in tissue homoeostasis.

Metformin has anticancer [4,5] and pro-longevity [6,7,8] effects in addition to its antidiabetic effect. However, its antidiabetic mechanism remains elusive. Metformin is thought to exert its primary antidiabetic action through suppression of hepatic glucose production. Subsequently, the primary mechanism of action has been suggested to be the inhibition of mitochondrial complex I [9,10]. Metformin is also reported to activate AMP-activated protein kinase (AMPK) through inhibition of mitochondrial complex I [10]. However, several studies have refuted the hypothesis of a direct action of metformin on complex I [11,12,13].

Emerging evidence suggests that metformin could modulate the functions of immune cells, especially macrophages. It has been recently demonstrated that metformin enhanced the anti-inflammatory properties of macrophages via AMPK-mediated modulation of M2 macrophage activation [14,15,16,17,18]. Alternatively, the anti-inflammatory mechanisms of metformin could be achieved by reducing the pro-inflammatory macrophages. Recently, several studies suggested that several anti-inflammatory drugs (such as curcumin and luteolin) were involved in regulation of macrophage activation via apoptosis of pro-inflammatory macrophages [19,20,21]. Apoptosis of pro-inflammatory macrophages promoted macrophage clearance and resolved inflammation [22]. Thus, we hypothesized that metformin could attenuate inflammatory responses by inducing apoptosis of pro-inflammatory macrophages.

The purpose of this study was to investigate the effect of metformin treatment on macrophage apoptosis and explore its underlying mechanism. Our results show that metformin promoted apoptosis of macrophages via G0S2 induction. In particular, metformin induced apoptosis of M1 macrophages, based on M1 marker genes in apoptotic macrophages. The results from this study have revealed the anti-inflammatory properties of metformin by possibly inducing apoptosis of inflammatory M1 macrophages, and provide a novel anti-inflammatory mechanism of metformin on M1 macrophages via induction of apoptosis.

## 2. Materials and Methods

### 2.1. Cell Culture and Metformin Treatment

The chicken HD11 macrophage cell line and mouse macrophage RAW264.7 cells were used for this study. HD11 cells were seeded into 6-well plates in Dulbecco’s modified Eagle’s medium (DMEM; Gibco, Waltham, MA, USA) with 10% fetal bovine serum (FBS) and cultured at 41 °C in 5% CO_2_ and 95% humidity, then treated with 1 mM, 5 mM and 25 mM metformin for 48 h. RAW264.7 cells were seeded into 6-well plates in DMEM with 10% FBS and cultured at 37 °C in 5% CO_2_ and 95% humidity, then treated with 1 mM and 5 mM metformin for 48 h.

### 2.2. Plasmids and Constructs

Full-length sequences of G0S2 were amplified from HD11 cDNAs with the following primers: forward primer 5′-CCCAAGCTTATGGAAACCATGCACG AGC-3′ and reverse primer 5′-CGCGGATCCTTAGGATGCATGCTGCCTG-3′, and then cloned into the eukaryotic expression vector pcDNA3.1.

### 2.3. DNA Transfection

HD11 cells and mouse macrophage RAW264.7 cells were plated on 6-well plates and transfected with pcDNA3.1-G0S2 or pcDNA3.1-EGFP expression plasmids using the Xfect™ Transfection Reagent (Takara, Japan) according to the manufacturer’s instructions. After 48 h transfection, cells were collected for RNA and protein analysis.

### 2.4. Cell Proliferation Assay

Cell viability was determined using the MTT assay following the manufacturer’s instructions (Beyotime, Shanghai, China). Briefly, 20 µL (50 mg/mL) MTT was added, and cells were incubated for an additional 2 h. The purple-blue MTT formazan precipitate was dissolved in DMSO. The activity of the mitochondria was evaluated by measuring the optical density at 570 nm. All MTT assays were performed in sextuplets and repeated in 3 independent experiments. CCK-8 assay was performed using the Cell Counting Kit-8 (Beyotime, China), following the manufacturer’s instructions.

### 2.5. Apoptotic DNA Ladder Detection

During apoptosis, activated nucleases degrade the higher order chromatin structure of DNA into fragments of 50 to 300 kilobases and subsequently into small DNA pieces of about 200 base pairs in length. These DNA fragments can be extracted from cells and easily visualized by agarose gel electrophoresis followed by ethidium bromide staining. Briefly, following treatment with metformin or DMSO for 48 h, cells were washed and the chromosomal DNA was extracted by the DNA Kit (AXYGEN, Union City, CA, USA). DNA fragmentation in apoptotic cells was detected by agarose gel electrophoresis.

### 2.6. Flow Cytometry

Apoptosis was analyzed by Annexin V Apoptosis Detection kit according to the manufacturer’s recommendations (BD Bioscience, East Rutherford, NJ, USA). FlowJo software (v. 10 TreeStar) was used for data analysis and graphic representation.

### 2.7. Library Construction for RNA-Seq

Total RNA was isolated from 1 mM metformin-treated chicken HD11 macrophages using the RNeasy mini kit (Qiagen, Hilden, Germany). Paired-end libraries were synthesized by using the TruSeq^®^ RNA Sample Preparation Kit (Illumina, San Diego, CA, USA) following the TruSeq^®^ RNA Sample Preparation Guide. Briefly, the poly-A containing mRNA molecules were purified using poly-T oligo-attached magnetic beads. Purified libraries were quantified by Qubit^®^ 2.0 Fluorometer (Life Technologies, Carlsbad, CA, USA) and validated by Agilent 2100 bioanalyzer (Agilent Technologies, USA) to confirm the insert size and calculate the mole concentration. Clusters were generated by cBot with the library diluted to 10 pM and then were sequenced on the Illumina HiSeq X-ten (Illumina, USA). The library construction and sequencing were performed at Shanghai Biotechnology Corporation.

### 2.8. Data Analysis for Gene Expression

Sequencing raw reads were pre-processed by filtering out rRNA reads, sequencing adapters, short-fragment reads and other low-quality reads. We used Hisat2 (version 2.0.4) [23] to map the cleaned reads to the chicken reference genome with 2 mismatches. After genome mapping, Stringtie (version 1.3.0) [24,25] was run with a reference annotation to generate FPKM values for known gene models. Differentially expressed genes were identified using edgeR [26]. The *p*-value significance threshold in multiple tests was set by the false discovery rate (FDR). The fold changes were also estimated according to the FPKM in each sample. The differentially expressed genes were selected using the following filter criteria: FDR ≤ 0.05 and fold change ≥2.

### 2.9. Gene Set Enrichment Analysis (GSEA)

Gene set enrichment analysis was performed by the GSEA software (version 4.0.3) using the different expression genes in metformin-treated macrophages. GSEA is a computational method that determines whether an a priori defined set of genes shows statistically significant, concordant differences between two biological states (e.g., phenotypes) [27,28].

### 2.10. Reverse Transcription and Quantitative PCR

Total RNA was first removed from the genomic DNA and then reverse-transcribed using the PrimeScript RT reagent Kit with gDNA Eraser (Takara, Japan) following the manufacturer’s instructions. The gDNA Eraser-treated RNA samples were reverse-transcribed with the RT mix primer at 37 °C for 15 min by PrimeScript^®^ Reverse Transcriptase (Takara, Japan). The quantitative PCR (qPCR) was performed with gene-specific primers and SYBR Green Master Mix (Takara, Japan) on the CFX Connect™ Real-Time PCR Detection System (Bio-Rad, Hercules, CA, USA); primers are listed in Table 1. *GAPDH* and *β-Actin* RNA levels were used as internal controls to normalize gene expression.

### 2.11. Statistical Analyses

The statistical analysis was performed using the Statistical Product and Service Solutions (version 16.0) software. Statistical significance was assessed using a two-tailed unpaired Student’s *t*-test with a *p* value threshold of <0.05.

## 3. Results

### 3.1. Metformin Induces Apoptosis in Macrophages

First, the effects of metformin on the proliferation/viability of chicken macrophages were measured by MTT (3-(4, 5-dimethylthiazolyl-2)-2, 5-diphenyltetrazolium bromide) and CCK-8 (Cell Counting Kit-8) assays. As shown in Figure 1A,B, metformin-treated chicken macrophages exhibited a lower cell viability rate compared to the control cells at 24 and 48 h. The formation of the DNA ladder in gel electrophoresis was obvious in 25 mM metformin-treated chicken macrophages (Figure 1C). The effect of metformin on apoptosis of chicken macrophages was further evaluated using flow cytometry based on propidium iodide (PI) and Annexin V-FITC staining. The percentages of apoptotic cells (Q2 and Q3 regions) were significantly increased in chicken macrophages treated with metformin at 5 mM and 25 mM (Figure 1D). Metformin also induces apoptosis in mouse macrophages. Compared to the control cells, metformin-treated mouse macrophages exhibited a lower cell viability rate at 48 h (Figure 1E,F), and the percentages of apoptotic cells were significantly increased in mouse macrophages treated with metformin at 1 mM and 5 mM (Figure 1G).

### 3.2. More M1 Marker Genes Were Detected in Metformin-Induced Apoptotic Macrophages

The apoptotic and nonapoptotic cells were further separated from metformin-treated macrophages to investigate whether metformin attenuated inflammatory responses by inducing apoptosis of pro-inflammatory macrophages (M1). In metformin-treated chicken macrophages, we found that M1 markers such as CD86, IL-6, IL-1β and NOS2 were higher in apoptotic macrophages compared to nonapoptotic cells (Figure 2A). However, metformin reduced M2 markers such as CD206, IL-4 and IL-10 in apoptotic macrophages (Figure 2B). In metformin-treated mouse macrophages, M1 markers such as CD86, IL-6, IL-1β and NOS2 in apoptotic macrophages were also higher (Figure 2C). On the contrary, expression of M2 markers such as CD206, Arg-1 and IL-10 were lower in apoptotic macrophages than in nonapoptotic cells (Figure 2D). These results indicated that metformin may attenuate inflammatory responses by inducing apoptosis of pro-inflammatory macrophages (M1).

### 3.3. Metformin Affects Gene Expression Profiles in Macrophages

To investigate which gene changes underlie the observed effects of metformin on the survival rates of macrophages, the gene expression of 1 mM metformin-treated HD11 cells was profiled by RNA-Seq. A total of 1315 significantly differentially expressed genes (all of them FC > 2 and q-value < 0.05) were obtained from metformin-treated macrophages (as shown in Figure 3A and Appendix A). Among them, 576 were upregulated and 739 downregulated (Figure 3B). We found that several interleukins (IL-1β and IL-8) and interleukin receptor genes (IL1r2, IL2rb, IL13ra1 and IL13ra2) and chemokine genes (Ccl5, Ccl20, Cx3cl1, Ccli9, Ccli7 and Ggcl1) were upregulated but several interferon genes and interferon-stimulated genes (Isgs), including Ifn-A, Ifn-B, Ifih1, Ifitm5 and Ifitm10 were downregulated. Metformin treatment also altered the expressions of cell growth related genes (including Tp73, Cdc23, Mcm3, Cpeb2, Adcy3 and Mapk10) and cell death related genes (including Rps6ka2, Fos, Bcl2, Tnfsf6 and Birc2) in macrophages. RNA-Seq data were validated by RT-qPCR analyses of 10 selected genes, including FGF8, FOS, IFN-β, SERPINF1, CCL20, CCL5, GGCL1, IL-1β, IL-8 and CX3CL1 (Figure 3C).

### 3.4. G0S2 Has the Most Significant and Positive Correlation with the Extrinsic Apoptotic Signaling Pathway in Metformin-Treated Macrophages

The gene set enrichment analysis further revealed that several pathways were significantly changed in metformin-treated macrophages. The top five significantly enriched pathways included regulation of exocytosis, the regulation of regulated secretory pathway, positive regulation of DNA binding transcript factor activity, regulation of RAS protein signal transduction and the extrinsic apoptotic signaling pathway (Figure 4 and Figure 5A).

Next, we focused on the extrinsic apoptotic signaling pathway to reveal the mechanism of apoptosis in macrophages induced by metformin. In the heat-map of this pathway, expression of the G0S2 (G0/G1 switch 2) gene was obviously increased, while expression of the apoptosis inhibitor Bcl-2 (B-cell lymphoma 2) was reduced in metformin-treated macrophages (Figure 5B). The leading edge analysis of the top five pathways performed by GSEA software confirmed that G0S2 was the most significantly and positively correlated with the extrinsic apoptotic signaling pathway (Figure 5C, blue arrow). Thus, we speculated that G0S2 might be a major player for apoptosis of metformin-treated macrophages.

### 3.5. Metformin Promotes Apoptosis via G0S2 Induction

To further understand the role of G0S2 in apoptosis induced by metformin on macrophages, quantitative PCR was performed for G0S2 and Bcl-2, an important downstream target gene for G0S2. Quantitative PCR confirmed the RNA-seq results that metformin altered these two gene expression levels in macrophages, with upregulation of *G0S2* and downregulation of *Bcl-2* (Figure 6A). Overexpression of G0S2 did not affect the mRNA expression levels of Bcl-2 gene in macrophages (Figure 6B). However, it significantly increased the mRNA expression levels of beclin-1, the direct target gene of Bcl-2 (Figure 6B). In addition, transfection of *G0S2* for 48 h in HD11 cells significantly induced apoptosis in a dose-dependent way (Figure 6C). These results suggested that G0S2 induction by metformin could induce apoptosis in macrophages and G0S2 exerted the pro-apoptotic effect at least in part through inhibition of Bcl-2 functions.

## 4. Discussion

Metformin has recently attracted attention as a new supportive therapeutic drug against a variety of diseases, including inflammation, cancer, anti-aging and infectious diseases. Anti-inflammatory effects have been mediated through macrophages. However, the anti-inflammatory mechanisms of metformin on macrophages are still not fully understood.

The present study has provided evidence for additional anti-inflammatory mechanisms of metformin’s actions on macrophages (Figure 7). It is well known that metformin can elicit anti-inflammatory effects by promoting M2 macrophage activation through induction of AMPK activation. There are at least three proposed mechanisms for AMPK-mediated modulation of macrophage activation by metformin. The first is inhibition of STAT3 activation via AMPK. STAT3 inhibited the phorbol myristate acetate (PMA)-induced macrophage activation, but this inhibition was removed by metformin-induced AMPK activation [17,18]. The second is related to inhibition of mTOR/NLRP3 inflammasome activity via AMPK. Activating and phosphorylating AMPK suppressed mTOR/NLRP3 inflammasome signaling pathway activation by reducing NLRP3, IL-1β and caspase-1 protein expression, which boosted M2 macrophage activation [15,29]. Finally, AMPK activation by metformin induced ATF-3 expression, resulting in a reduction of IL-6 and TNF-α in lipopolysaccharide (LPS)-induced inflammatory macrophages [30]. Our results support a novel anti-inflammatory mechanism of metformin on macrophages by inducing apoptosis of M1 macrophages through activation of G0S2 and inhibition of Bcl-2 (Figure 7).

Metformin induced the apoptosis of M1 phenotype macrophages in this study. It has been demonstrated that metformin induced endoplasmic reticulum stress [31,32], leading to apoptosis in M1, but not M2 macrophages. The transient receptor potential canonical 3 (TRPC3) channel contributes to endoplasmic reticulum stress-induced apoptosis in M1, but not in M2 macrophages. TRPC3-deficient macrophages polarized to the M1 phenotype showed reduced apoptosis [33,34]. In addition, the human cytolytic fusion proteins (CFP) also specifically eliminated polarized M1 macrophages in a transgenic mouse model of cutaneous chronic inflammation [35]. How metformin is related to TRPC3 and CFP needs further investigation.

Our proposed mechanism is also supported by other research findings from other anti-inflammatory drugs (such as curcumin and luteolin). These drugs could regulate macrophage activation via apoptosis [19,20,21]. Bcl-2 is a crucial anti-apoptotic gene and central regulator of caspase activation and cellular life-or-death switch [36,37,38]. Inhibition of Bcl-2 could lead to mitochondrial apoptosis in macrophages, and this intrinsic apoptosis would promote activation of IL-1β [39]. This may explain why metformin treatment activates IL-1β in certain macrophages. G0S2 specifically interacts with Bcl-2 and promotes apoptosis by preventing the formation of protective Bcl-2/Bax heterodimers [40]. Overexpression of G0S2 in this study significantly induced apoptosis in macrophages. Therefore, induction of apoptosis by G0S2 upregulation and Bcl-2 inhibition is an important anti-inflammatory mechanism of metformin on macrophages.

An important finding from this study is that metformin promoted apoptosis of macrophages via G0S2 induction. This new finding has added an additional layer of anti-inflammatory properties of metformin by possibly inducing apoptosis of inflammatory M1 macrophages, besides inducing alternative activation on non-apoptotic macrophages. Recently, it also has been suggested that the apoptotic cells may drive alternative activation of non-apoptotic macrophages [41]. All of these results support the notion that metformin has anti-inflammatory properties.

Induction of macrophage apoptosis by metformin was concentration-dependent in this study. The common concentrations of metformin for in vitro studies range between 0.5 and 10 mM. Compared to mouse macrophages, a higher concentration of metformin was required in chicken macrophages to induce a strong apoptosis. Nevertheless, treatment of both chicken macrophages and mouse macrophages with a low concentration of 1 mM metformin led to apoptosis in macrophages. However, whether the present results can be applied to in vivo application needs further investigation.

## 5. Conclusions

In conclusion, our results provide a novel anti-inflammatory mechanism of metformin on macrophages. Namely, it induces apoptosis of macrophages via G0S2 induction. In particular, metformin induced apoptosis of M1 macrophages, based on M1 marker genes in apoptotic macrophages. These findings may be significantly important for understanding the mechanisms of metformin’s anti-inflammatory actions.

## Figures and Tables

**Figure 1 genes-12-01437-f001:**
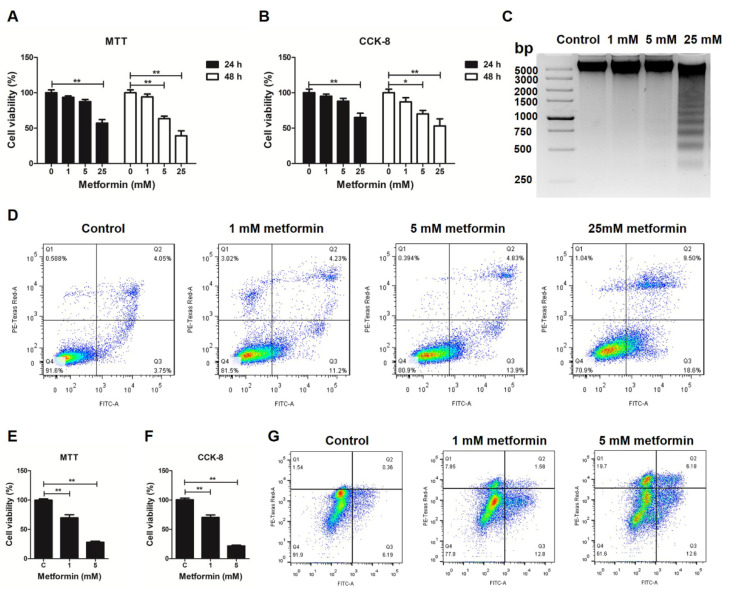
The influence of metformin on cell proliferation and apoptosis in macrophages. MTT (**A**) and CCK-8 (**B**) assays were performed to analyze the cell proliferation of HD11 cells treated with metformin for 24 and 48 h. (**C**) Apoptotic DNA ladder detection for HD11 cells treated with metformin for 48 h. (**D**) Metformin-induced apoptosis in HD11 cells by flow cytometry. The X axis represents Annexin V-FITC, and the Y axis represents propidium iodide (PI). MTT (**E**) and CCK-8 (**F**) assays were performed to analyze the cell proliferation of RAW264.7 cells treated with metformin for 48 h. (**G**) Metformin-induced apoptosis in RAW264.7 cells by flow cytometry. The X axis represents Annexin V-FITC, and the Y axis represents PI. Q1 represents necrotic cells, Q2 represents late apoptotic cells, Q3 represents early apoptotic cells, and Q4 represents normal cells. * represents *p* < 0.05 and ** represents *p* < 0.01.

**Figure 2 genes-12-01437-f002:**
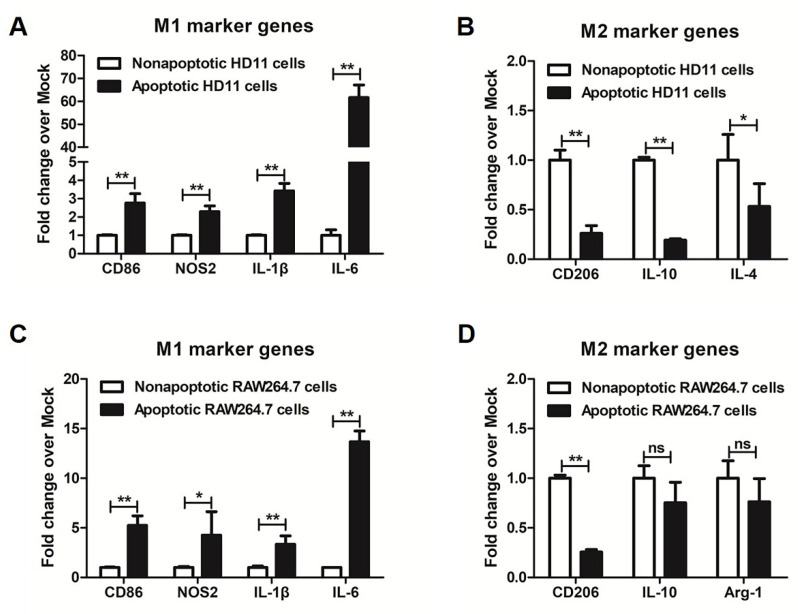
The influence of metformin on M1 and M2 marker genes in apoptotic macrophages. RT-qPCR analysis of M1 (**A**) and M2 (**B**) marker genes in apoptotic and nonapoptotic chicken macrophages separated from 25 mM metformin-treated chicken HD11 cells for 48 h. RT-qPCR analysis of M1 (**C**) and M2 (**D**) marker genes in apoptotic and nonapoptotic macrophages separated from 5 mM metformin-treated mouse RAW264.7 cells for 48 h. * represents *p* < 0.05 and ** represents *p* < 0.01.

**Figure 3 genes-12-01437-f003:**
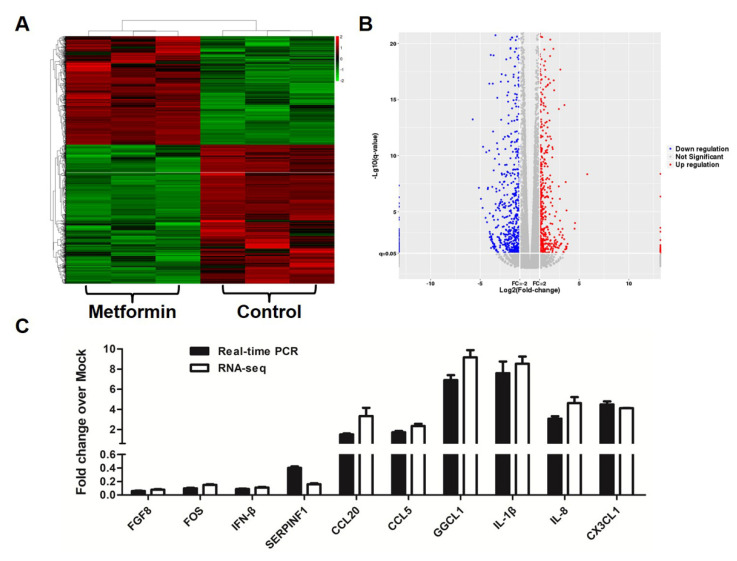
RNA-seq analysis of metformin responsive-genes in macrophages. (**A**) Heat-map expression profiles of metformin responsive-genes in macrophages. (**B**) Volcano plot of the p values as a function of weighted fold change for mRNAs in metformin-treated macrophages. The X axis represents Log2 (fold change) and the Y axis represents −Log10 (q-values). (**C**) Validation of RNA-Seq data for 10 randomly selected differentially expressed genes by real-time PCR.

**Figure 4 genes-12-01437-f004:**
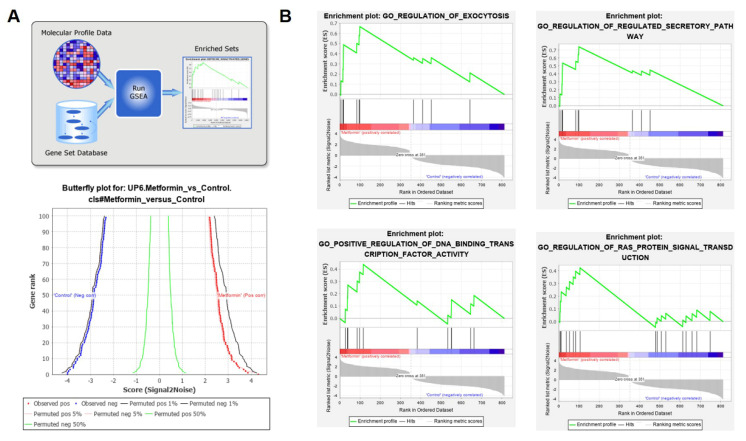
Gene set enrichment analysis (GSEA) of metformin responsive-genes in macrophages. (**A**) The work flow chart of GSEA and butterfly plot for metformin responsive-genes in macrophages. (**B**) Enrichment plots for regulation of exocytosis, regulation of regulated secretory pathway, positive regulation of DNA binding transcript factor activity and regulation of RAS protein signal transduction in the data set. The plots contain the enrichment score (ES) and positions of GeneSet members on the rank-ordered list. The X axis represents the rank-ordered dataset and the Y axis represents the enrichment score (ES).

**Figure 5 genes-12-01437-f005:**
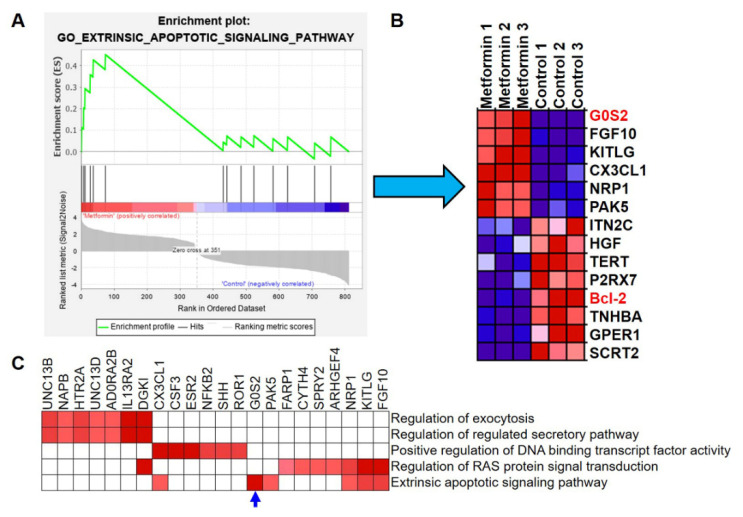
G0S2 is the most significant and positive correlated protein with the extrinsic apoptotic signaling pathway in metformin-treated macrophages. (**A**) Enrichment plot of the extrinsic apoptotic signaling pathway in the data set, including the ES score and positions of GeneSet members on the rank-ordered list. (**B**) Heat map of the analyzed GeneSet in the extrinsic apoptotic signaling pathway. (**C**) Heat map of top five pathways clustered leading edge subsets. Rows are gene sets and columns are genes. This matrix is clustered.

**Figure 6 genes-12-01437-f006:**
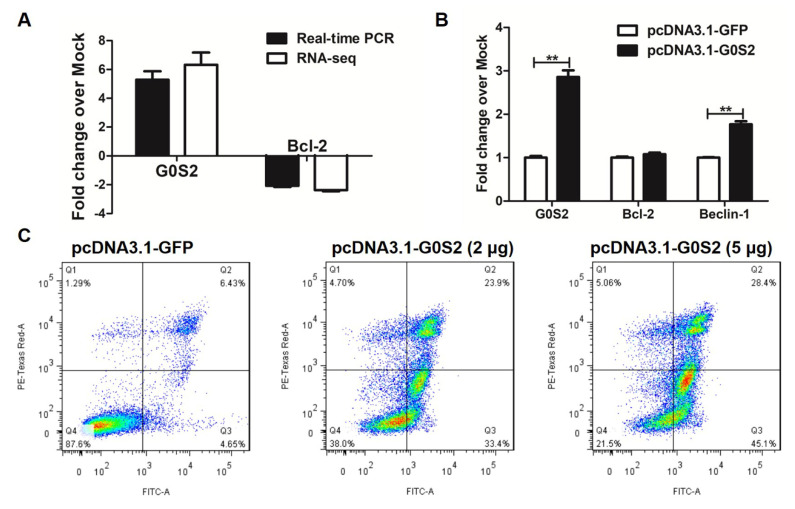
Metformin modulates apoptosis via G0S2 in macrophages. (**A**) Validation of RNA-Seq data for G0S2 and Bcl-2 in metformin-treated HD11 cells by real-time PCR. (**B**) RT-qPCR analysis of G0S2, Bcl-2 and Beclin-1 gene expression in macrophages transfected with 2-μg G0S2 plasmid. (**C**) Flow cytometry results showed that over-expression of G0S2 induced apoptosis in HD11 cells. The X axis represents Annexin V-FITC and the Y axis represents PI. Q1 represents necrotic cells, Q2 represents late apoptotic cells, Q3 represents early apoptotic cells and Q4 represents normal cells. ** represents *p* < 0.01.

**Figure 7 genes-12-01437-f007:**
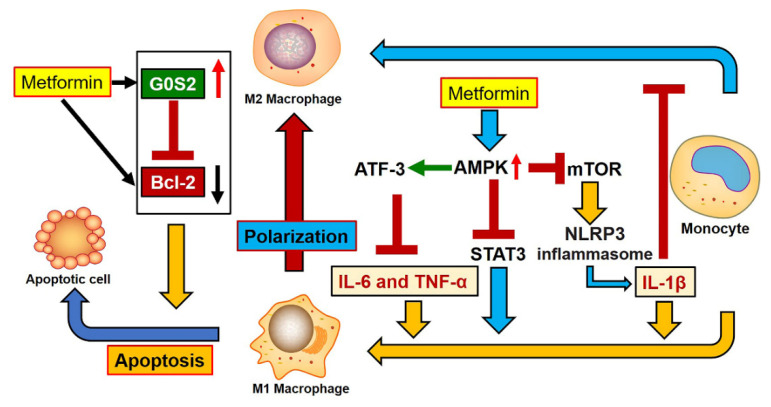
A working model of the anti-inflammatory mechanism for metformin on macrophages. The model describes how metformin promotes M2 macrophage activation through induction of AMPK activation, as previously proposed [15,17,18]. Synergistically, the results from this study support a novel anti-inflammatory mechanism of metformin on macrophages by inducing apoptosis of M1 macrophages through activation of G0S2 and inhibition of Bcl-2.

**Table 1 genes-12-01437-t001:** Primers used in this study.

Primer Name	Nucleotide Sequence 5′-3′
Chicken CD86 fwd	GGATGTCTTACAGGATGCT
Chicken CD86 rev	CTGCTCTCCAAGGTGAAG
Chicken NOS2 fwd	CCACTCATTCTCCAAGCAA
Chicken NOS2 rev	AGGCAGAGCATACCACTT
Chicken IL-6 fwd	GTCGAGTCTCTGTGCTAC
Chicken IL-6 rev	CTTCAGATTGGCGAGGAG
Chicken CD206 fwd	GAGGACTGCGTTGTTATGA
Chicken CD206 rev	TCTTCTGTCGGTGCTTCT
Chicken IL-10 fwd	GCTCTGAACTGCTGGATG
Chicken IL-10 rev	ATGCTCTGCTGATGACTG
Chicken IL-4 fwd	AATGACATCCAGGGAGAGG
Chicken IL-4 rev	CAGGTTCTTGTGGCAGTG
Chicken CCL5 fwd	GCTCTGTCCCTCTCCATCCT
Chicken CCL5 rev	GTTGAAGCAGCACACGGTTG
Chicken IL-1β fwd	TAGATGTCGTGTGTGATGAG
Chicken IL-1β rev	GTAGAAGATGAAGCGGGTC
Chicken IL-8 fwd	ACGCTGGTAAAGATGGGGAA
Chicken IL-8 rev	GCACACCTCTCTTCCATCC
Chicken CX3CL1 fwd	GGTGGAGAAGATCGTGAAG
Chicken CX3CL1 rev	CTGGAGGTGAAGGTGGTA
Chicken CCL20 fwd	TCAAGAGGATGTCAATGTGA
Chicken CCL20 rev	AGAGATAGTGGTGAGTAAGC
Chicken GGCL1 fwd	CGAGCAAGGTGATGATGTA
Chicken GGCL1 rev	TTGGCACAGCACTTCTTC
Chicken IFN-β fwd	GCCCACACACTCCAAAACACTG
Chicken IFN-β rev	TTGATGCTGAGGTGAGCGTTG
Chicken FGF8 fwd	ACTGATCGGCAAGAGTAAC
Chicken FGF8 rev	CTCGTACTTGGCGTTCTG
Chicken FOS fwd	CTACTGTGTTCCTGGCAAT
Chicken FOS rev	ACATTCAGACCACCTCAAC
Chicken SERPINF1 fwd	CCACAGCCAACTAGAGAAG
Chicken SERPINF1 rev	AGCAAGGAGAATACTGACATC
Chicken Bcl-2 fwd	AGACCTACCTGCTTACACT
Chicken Bcl-2 rev	GCTTACTCTGACTGCTCTC
Chicken G0S2 fwd	TCAGCCAGAAGCCCAACAGG
Chicken G0S2 rev	GATGACCACGCCGAAGAACG
ALVJ env fwd	TGCGTGCGTGGTATTATTTC
ALVJ env rev	AATGGTGAGGTCGCTGACTGT
Chicken β-Actin fwd	GAGAAATTGTGCGTGACATCA
Chicken β-Actin rev	CCTGAACCTCTCATTGCCA
Chicken GAPDH fwd	GAGAAACCAGCCAAGTATGA
Chicken GAPDH rev	CTGGTCCTCTGTGTATCCTA
Mouse CD86 fwd	GACTCTACGACTTCACAATG
Mouse CD86 rev	TGTTAATGTCTGTTGGAGGA
Mouse NOS2 fwd	TACTGCTGGTGGTGACAA
Mouse NOS2 rev	CTGAAGGTGTGGTTGAGTT
Mouse IL-6 fwd	GCCAGAGTCCTTCAGAGA
Mouse IL-6 rev	GATGGTCTTGGTCCTTAGC
Mouse IL-1β fwd	CTACAGGCTCCGAGATGA
Mouse IL-1β rev	CGTTGCTTGGTTCTCCTT
Mouse CD206 fwd	GGCAAGTATCCACAGCAT
Mouse CD206 rev	GGTTCCATCACTCCACTC
Mouse IL-10 fwd	GGTTGCCAAGCCTTATCG
Mouse IL-10 rev	TCCACTGCCTTGCTCTTA
Mouse Arg1 fwd	GCAGAGGTCCAGAAGAATG
Mouse Arg1 rev	GGAGTGTTGATGTCAGTGT
Mouse GAPDH fwd	GTGAAGGTCGGTGTGAAC
Mouse GAPDH rev	CTTGACTGTGCCGTTGAA

## Data Availability

All the data mentioned in this paper are available in the article and Appendix A.

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
