# Peer review of "Metformin Triggers Apoptosis and Induction of the G0/G1 Switch 2 Gene in Macrophages"

_genes, 2021, doi:10.3390/genes12091437_

Round 1

Reviewer 1 Report

This version of the paper “Metformin triggers apoptosis and induction of the 2 G0/G1 Switch 2 gene in macrophages” by Hu et al is much improved.

For a better understanding of the paper’s message, I think the authors should try to clarify the interpretation of their observations. The observation is that Metformin promoted apoptosis of macrophages via G0S2 induction. The interpretation is therefore that Metformin could be having an additional layer of anti-inflammatory properties by inducing apoptosis of inflammatory macrophages, besides inducing alternative activation on non-apoptotic macrophages. It is also possible that what drives alternative activation of macrophages is the apoptotic cells and not Metmorfin itself (Determining the effector response to cell death. Rothlin CV, Hille TD, Ghosh S. Nat Rev Immunol. 2021). This is important throughout all the sections of the manuscript: abstract, introduction, results, discussion, and conclusion.

In this regard, I wonder if the authors could show that Metformin is selectively triggering apoptosis on M1 macrophages? To do this, they could repeat experiments as in Figure 1C and measure an M1 marker as well. One option could be Il1b or Tnf gene by PrimeFlow? Alternatively, they could treat macrophages with and without IL-4 (recombinant chicken available in Abcam and other companies) and LPS (for 24h for example) and then treat each group with and without Metformin to see if the pro-apoptotic effect of Metformin is stronger when macrophages are classically activated vs alternatively activated macrophages.

Minor comments

• Avoid using the term polarisation, I believe activation is a more suitable term

• The paragraph discussing the relevance of chicken macrophages as a model to study human diseases is too long and distracting and should be reduced as much as possible.

Reviewer 2 Report

The study provides evidence for the mechanism of metformin in inhibiting monocyte/macrophages from pro-inflammatory M1 responses in type 2 diabetic chicken and mouse diabetic models.  However, the descriptions of the results often need to be increased to help the readers understand the results in the figures.  For example the first sentence in the results indicates that panels A and B of figure 1 show lower viability of treatments, but does not explain what MTT and CCK-8 are or why they were chosen for analysis.  

(l. 178) states that IL-10 and Arg-1 were lower, but the figure does not indicate significance.  The text could indicate that it is a trend.  Similarly in panel B, IL-4 is not indicated as significant.  Its p value could be given, which is likely to be close to significance.

(l. 199)  The list of 11 genes should be separated to show the growth vs death genes and whether they split from blue to red.

(l. 201-202)  The first 4 are significantly inhibited and the next 6 significantly increased.  How do they relate to the model?

Edits

(l. 54)  ‘achieved’

(l. 67)  ‘on M1 macrophages’

(l. 70)  ‘macrophage RAW26.7 cells’

(l. 78)  ‘G0S2 were amplified’

(l. 104)  ‘Apoptosis was analyzed’

(l. 115)  ‘Clusters were generated’

(l. 135)  ‘RNA was first removed from’

(l. 141)  ‘primers are listed’

(l. 153)  It would help if the Flow cytometry results in the figure were explained further. What areas of change determine the apoptosis?

(l. 162 and 165)  ‘assays were’

(l. 165)  Define PI

(l. 173 and 176)  ‘markers’

(l. 178)  ‘were lower’

(l. 185)  ’25 mM’

(l. 187)  ‘5 mM’

(l. 198)  ‘expressions of’ 

(l. 208 and elsewhere in the next 2 paragraphs)  ‘the …. pathway’

(. 221)  ‘(Figure 5C, blue arrow)’

(l. 231)  ‘correlated protein with’

(l. 263)  ‘effects by promoting’

(l. 279)  ‘The model describes how’  There is no black dotted frame.

(l. 319)  ‘macrophages. Namely, it induces apoptosis’

Reviewer 3 Report

In this manuscript, the authors showed metformin induced apoptosis preferentially in M1 macrophages, explaining its anti-inflammatory effect. Using RNA-seq and GSE analysis, they successfully identified G0S2 and Bcl-2 as up- and down-regulated genes by metformin treatment, respectively. In general, the experiments are logical and the results are interesting.

1) My concern is whether these results can be applied in vivo. In clinical setting, plasma concentration of metformin is usually raised only up to 1 μg/mL when 1,000 mg metformin is orally administered. Please explain why high concentration was required in in vitro study and discuss this issue in the manuscript.

2) Can this apoptotic effect of metformin be observed on the other type of cells such as cancer cells? It would be appreciated if the authors make a comment on this.

Minor comment

It is hard to read the vertical and horizontal axes in figs 1D, 1G, 3B, 4.

Round 2

Reviewer 1 Report

The authors have addressed all my concerns. 

This manuscript is a resubmission of an earlier submission. The following is a list of the peer review reports and author responses from that submission.

Round 1

Reviewer 1 Report

In the manuscript entitled “Metformin triggers apoptosis and induction of the 2 G0/G1 Switch 2 gene in macrophages” Hu et al. unravel a previously unappreciated apoptotic effect of Metformin in macrophages. While the experiments are well designed and the manuscript is well written, there seems to be a disconnection between their observations and the previously published effects of Metformin on macrophages. When reading the manuscript, one gets the impression that the authors assume that the apoptotic effects of Metformin on macrophages and the previously published immunomodulatory effects on their phenotype occur in the same cell. However, this might not be the case. For example, it is well known that efferocytosis can induce a pro-resolution phenotype on macrophages and can synergise with IL-4 to induce tissue repair (Bosurgi et al., 2017). Given this, it is conceivable to think that Metformin is moderately toxic to macrophages and induces apoptosis in a portion of cells. Subsequently, the removal of cell corpses by sister cells is what induces the pro-resolving phenotype, and not Metformin directly. Therefore, I suggest the authors follow the induction of apoptosis and inflammasome in parallel to changes in macrophage phenotype (classic vs. alternative activation markers) in a time course experiment. It would be perhaps convenient to separate apoptotic vs non-apoptotic cells in this experiment before analysis. If there is evidence for the apoptotic process and the phenotype change to occur in different cells, I suggest the authors try to block efferocytosis to show to show that the change in phenotype is an indirect effect of Metformin. Finally, I suggest that crucial experiments should be repeated in primary macrophages such as BMDMs.

In terms of nomenclature, I suggest the authors follow the suggestions by Murray et al. (Murray et al., 2014) and replace M1 and M2 terminology for something more accurate. Also, the common misconception that M2 macrophages are anti-inflammatory needs to be corrected. These cells can be inflammatory and induce recruitment of eosinophils for instance. Despite this they do have pro-resolving capabilities.

Finally, figure 1C and 5C (FACS plots) are missing the labels of the axes. Are they Annexin V vs. PI?

References

Bosurgi, L., Cao, Y.G., Cabeza-Cabrerizo, M., Tucci, A., Hughes, L.D., Kong, Y., Weinstein, J.S., Licona-Limon, P., Schmid, E.T., Pelorosso, F., et al. (2017). Macrophage function in tissue repair and remodeling requires IL-4 or IL-13 with apoptotic cells. Science 356, 1072-1076.

Murray, P.J., Allen, J.E., Biswas, S.K., Fisher, E.A., Gilroy, D.W., Goerdt, S., Gordon, S., Hamilton, J.A., Ivashkiv, L.B., Lawrence, T., et al. (2014). Macrophage activation and polarization: nomenclature and experimental guidelines. Immunity 41, 14-20.

Author Response

Response to Reviewer 1 Comments

Point 1: In the manuscript entitled “Metformin triggers apoptosis and induction of the 2 G0/G1 Switch 2 gene in macrophages” Hu et al. unravel a previously unappreciated apoptotic effect of Metformin in macrophages. While the experiments are well designed and the manuscript is well written, there seems to be a disconnection between their observations and the previously published effects of Metformin on macrophages. When reading the manuscript, one gets the impression that the authors assume that the apoptotic effects of Metformin on macrophages and the previously published immunomodulatory effects on their phenotype occur in the same cell. However, this might not be the case. For example, it is well known that efferocytosis can induce a pro-resolution phenotype on macrophages and can synergise with IL-4 to induce tissue repair (Bosurgi et al., 2017). Given this, it is conceivable to think that Metformin is moderately toxic to macrophages and induces apoptosis in a portion of cells. Subsequently, the removal of cell corpses by sister cells is what induces the pro-resolving phenotype, and not Metformin directly. Therefore, I suggest the authors follow the induction of apoptosis and inflammasome in parallel to changes in macrophage phenotype (classic vs. alternative activation markers) in a time course experiment. It would be perhaps convenient to separate apoptotic vs non-apoptotic cells in this experiment before analysis. If there is evidence for the apoptotic process and the phenotype change to occur in different cells, I suggest the authors try to block efferocytosis to show to show that the change in phenotype is an indirect effect of Metformin. Finally, I suggest that crucial experiments should be repeated in primary macrophages such as BMDMs. 

Response 1: Thanks for this comment. We have performed the proliferation assay of macrophages in the presence of different concentrations of metformin in a time course experiment. We found that metformin was moderately toxic to macrophages and induced apoptosis in a portion of cells. The proliferation results were confirmed by the Flow cytometry analysis. We are not sure whether a difference of the apoptosis process occurs in different phenotype macrophages (the pro-inflammatory “classically” activated and the “alternatively” activated). As the reviewer rightly points out, our focus for this manuscript is to unravel a previously unappreciated apoptotic effect and we did not reveal whether this effect is direct or indirect. However, we will consider perform these experiments in future research.

Point 2: In terms of nomenclature, I suggest the authors follow the suggestions by Murray et al. (Murray et al., 2014) and replace M1 and M2 terminology for something more accurate. Also, the common misconception that M2 macrophages are anti-inflammatory needs to be corrected. These cells can be inflammatory and induce recruitment of eosinophils for instance. Despite this they do have pro-resolving capabilities.

Response 2: Following your advice, we have corrected them in the revised manuscript as follows.

“…termed the pro-inflammatory M1 and anti-inflammatory M2” was modified as “…termed the pro-inflammatory “classically” activated (M1 macrophage) and the “alternatively” activated (M2 macrophage)”

“the anti-inflammatory M2 macrophage” was modified as “the M2 macrophage”

“STAT3 inhibited the M2 macrophage polarization” was modified as “STAT3 inhibited the phorbol myristate acetate (PMA)-induced macrophage polarization”

Point 3: Finally, figure 1C and 5C (FACS plots) are missing the labels of the axes. Are they Annexin V vs. PI?

Response 3: Thanks for this comment. The labels of the axes have been added in the legends for Figure 1C and 5C in the revised manuscript. The X axis represents Annexin V-FITC and the Y axis represents PI.

Reviewer 2 Report

Specific comments:

Rows 40-54 – The AMPK activation mediated by metformin is a consequence of mitochondrial complex I inhibition. Moreover, evidence is emerging today that, apart from the effect on complex I, metformin inhibits several other cellular processes, including DNA methylation, which may explain metformin’s antiproliferative effects. Thus, the authors should focus the introduction not exclusively on AMPK, but should extend their view on the broader effects of this drug, at least on complex I inhibition, if not on other mechanisms that are emerged recently.

Rows 67-71 – More than one macrophage model should be used, the data obtained by the authors should be tested in at least 2 other models (induced from THP1 or RAW monoctes; or extracted from human peripheral blood buffy coat or from mouse intraperitoneal cavity).

Rows 81-86 – Proliferation assay other than MTT should be performed, as this assay depends on mitochondrial respiration rate and metformin is a known complex I inhibitor.

Row 163 – The authors should specify which conditions were used to compare metformin treated versus untreated chicken macrophages. Metformin has shown to induce cytostatic antiproliferative effects at low doses and cytotoxic effects when used at high concentrations. If the 25mM metformin at 48 hours was used for differential expression analysis, then the apoptosis may be due to cytotoxic effects due to high dosage. The upregulation of exocytosis and secretory pathways is in line with over-dosing with the drug.

Row 225 – I do not understand how an anticancer effect may be observed by using only a macrophage model. If the authors wish to investigate the protumorigenic and antitumorigenic effects of macrophages, co-culture experiments should be set up.

General comments:

  1. No experiments were performed to investigate macrophage polarisation.
  2. Macrophages may promote both inflammatory and wound healing functions. The fact that metformin causes their apoptosis may have double-edged consequences on cancer, which has not been tackled at all in this manuscript.
  3. A set of randomly chosen differentially expressed genes should be tested by qRTPCR to validate the RNAseq data.
  4. Apart from overexpression of G0S2, the gene should be also inhibited (for example by silencing) to provide a proof of concept.
  5. The literature reports that in vitro macrophage models are affected by metformin in a way that it causes M2 polarization. However, a systemic treatment of animal cancer models with metformin has mainly been associated with M1 polarization. The author’s should in some way validate their results in vivo.
  6. No controls have been included in the experiments, such as AMPK western blot or oxygen consumption rate with Seahorse to ascertain metformin's effect on AMPK signalling and cell respiration.

Author Response

Response to Reviewer 2 Comments

Specific comments:

Point 1: Rows 40-54 – The AMPK activation mediated by metformin is a consequence of mitochondrial complex I inhibition. Moreover, evidence is emerging today that, apart from the effect on complex I, metformin inhibits several other cellular processes, including DNA methylation, which may explain metformin’s antiproliferative effects. Thus, the authors should focus the introduction not exclusively on AMPK, but should extend their view on the broader effects of this drug, at least on complex I inhibition, if not on other mechanisms that are emerged recently.

Response 1: Thanks for this comment. In the revised manuscript, we extend these view in the introduction section as follows:

“Metformin has anticancer [4, 5], prolongevity [6-8] effects in addition to its antidiabetic effect.  However, its mechanism of action remains elusive. Metformin is thought to exert its primary antidiabetic action through suppression of hepatic glucose production. Subsequently, the primary mechanism of action has been suggested to be the inhibition of mitochondrial complex I [9, 10]. Metformin was also reported to activate AMP-activated protein kinase (AMPK) through inhibition of mitochondrial complex I by the drug [10]. However, several studies have refuted the hypothesis of a direct action of metformin on complex I [11-13].

Emerging evidence suggests that metformin could modulate the functions of immune cells, especially macrophage. It has been recently demonstrated that metformin…”

Point 2: Rows 67-71 – More than one macrophage model should be used, the data obtained by the authors should be tested in at least 2 other models (induced from THP1 or RAW monoctes; or extracted from human peripheral blood buffy coat or from mouse intraperitoneal cavity).

Response 2: Thanks for this comment. Most of studies on the effects of metformin on macrophage only used one macrophage model (either human macrophages or mouse macrophages). Like humans or mice, chickens have been an important experimental system for developmental biology, immunology, and microbiology, leading to many fundamental discoveries (Brown, Hubbard et al. 2003). Of course, we will consider the human and mouse macrophage model in future research.

Reference:

Brown, W. R., et al. (2003). "The chicken as a model for large-scale analysis of vertebrate gene function." Nat Rev Genet 4(2): 87-98.

Point 3: Rows 81-86 – Proliferation assay other than MTT should be performed, as this assay depends on mitochondrial respiration rate and metformin is a known complex I inhibitor.

Response 3: Thanks for this comment. We did use both MTT and CCK-8 for proliferation assay, even though we only presented MTT results in the original submission since similar results were obtained by these two methods. Following your comment, we have added the results for CCK-8 in the revised manuscript.

Point 4: Row 163 – The authors should specify which conditions were used to compare metformin treated versus untreated chicken macrophages. Metformin has shown to induce cytostatic antiproliferative effects at low doses and cytotoxic effects when used at high concentrations. If the 25mM metformin at 48 hours was used for differential expression analysis, then the apoptosis may be due to cytotoxic effects due to high dosage. The upregulation of exocytosis and secretory pathways is in line with over-dosing with the drug.

Response 4: Thanks for this comment. According to the proliferation assay results, 1 mM metformin-treated macrophages owing to the low cytotoxic effects were used for RNA-seq analysis to compare metformin treated versus untreated chicken macrophages. This specify conditions has been stated in the revised manuscript as follows:

In the Materials and Methods section

“Total RNA was isolated from 1 mM metformin-treated chicken macrophages using RNeasy mini kit”

In the Results section

“…the gene expression of 1 mM metformin-treated HD11 cells was profiled by RNA-Seq”

Point 5: I do not understand how an anticancer effect may be observed by using only a macrophage model. If the authors wish to investigate the protumorigenic and antitumorigenic effects of macrophages, co-culture experiments should be set up.

Response 5: Thanks for this comment. Recent studies have revealed that anticancer effects of metformin mediated by oncoprotein MYC inhibition, and that the expression of MYC can be suppressed by the tumor suppressor G0S2. In our study, we further observed that metformin inhibited MYC through upregulation of G0S2. Therefore, we discussed this. However, we did not investigate the protumorigenic and antitumorigenic effects of macrophages. Thus, the statement “To further investigate how metformin elicited the anti-cancer effect through G0S2,” has been removed in the revised manuscript.

General comments:

Point 1: No experiments were performed to investigate macrophage polarisation.

Response 1: Thanks for this comment. The purpose of this study was to investigate the effect of metformin treatment on macrophage apoptosis and explore the involvement of underlying pathways. It is true that we did not investigate macrophage polarisation.

Point 2: Macrophages may promote both inflammatory and wound healing functions. The fact that metformin causes their apoptosis may have double-edged consequences on cancer, which has not been tackled at all in this manuscript.

Response 2: We fully agree with the reviewer that macrophages have double-edged consequences on cancer, e.g., macrophages are polarized into the cancer-inhibiting M1 and cancer-promoting M2 types in the tumour microenvironment. The effect of metformin on macrophages may be related to this specific conditions. The purpose of this current study was not to study the anti-cancer mechanism of metformin employed by a macrophage model. We have removed the sentence on anti-cancer effect by metformin.

Point 3: A set of randomly chosen differentially expressed genes should be tested by qRT-PCR to validate the RNAseq data.

Response 3: Thanks for this comment. We have randomly chosen 10 differentially expressed genes used for testing by qRT-PCR to validate the RNAseq data. Please see Figure 2C.

Point 4: Apart from overexpression of G0S2, the gene should be also inhibited (for example by silencing) to provide a proof of concept.

Response 4: It is true that we did not inhibit G0S2 gene expression in the current study. Unfortunately, we cannot perform additional experiment at this time and will consider this good suggestion in future research.

Point 5: The literature reports that in vitro macrophage models are affected by metformin in a way that it causes M2 polarization. However, a systemic treatment of animal cancer models with metformin has mainly been associated with M1 polarization. The author’s should in some way validate their results in vivo.

Response 5: Thanks for this comment. It is true that there are inconsistent experimental results between in vivo and in vitro macrophage models. Unfortunately, we cannot perform additional experiment in vivo at this time and we will consider this good suggestion in future research.

Point 6: No controls have been included in the experiments, such as AMPK western blot or oxygen consumption rate with Seahorse to ascertain metformin's effect on AMPK signalling and cell respiration.

Response 6: Thanks for this comment. In our experiments, macrophages treated without metformin were used as the control to determine the effect of metformin on macrophage apoptosis and explore its underlying mechanism. As we discussed before, whether metformin activates AMPK is still a matter of debate (the references has attached). In addition, the purpose of this study was to investigate the effect of metformin treatment on macrophages apoptosis and explore its underlying mechanism. Therefore, it may be not be necessary to ascertain metformin's effect on AMPK signalling and cell respiration.

References

Foretz, M., et al. (2019). "Understanding the glucoregulatory mechanisms of metformin in type 2 diabetes mellitus." Nat Rev Endocrinol 15(10): 569-589.

Wang, Y., et al. (2019). "Metformin Improves Mitochondrial Respiratory Activity through Activation of AMPK." Cell Rep 29(6): 1511-1523 e1515.

Martin-Rodriguez, S., et al. (2020). "Mitochondrial Complex I Inhibition by Metformin: Drug-Exercise Interactions." Trends Endocrinol Metab 31(4): 269-271

Cameron, A. R., et al. (2018). "Metformin selectively targets redox control of complex I energy transduction." Redox Biol 14: 187-197.

Fontaine, E. (2018). "Metformin-Induced Mitochondrial Complex I Inhibition: Facts, Uncertainties, and Consequences." Front Endocrinol (Lausanne) 9: 753.

Round 2

Reviewer 1 Report

My feeling is that the scope of the study in its present form, is too narrow and it doesn’t add much knowledge to the mechanism of action Metformin, if anything it adds confusion. As stated by the authors, macrophages can be beneficial and detrimental in any given disease depending on their activation status. Previous studies (mentioned in the introduction of the article) argue for a beneficial effect of Metformin during type 2 diabetes through the enhancement of the alternative activation of macrophages and/or suppression of their pro-inflammatory activities. Now the authors show that Metformin induces apoptosis in macrophages; but wouldn’t this be counterintuitive in terms of explaining the drug’s mode of action if it is killing the cells that mediate the therapeutic effects? Presumably, Metformin will be killing the M1s and enhancing (directly or indirectly) the M2 phenotype as I discussed in my previous review. In fact, this is what the authors hypothesise at the end of the introduction and yet fail to demonstrate: Thus, we hypothesized that metformin could attenuate inflammatory responses by inducing apoptosis of the inflammatory macrophages. This is precisely why apoptosis and activation of Metformin-treated macrophages should have been studied in parallel at the single cell level simply by flow cytometry for instance and not only in bulk analysis. This extra layer of understanding will broaden the article’s interest and impact.  

Author Response

Response to Reviewer 1 Comments

Point 1: My feeling is that the scope of the study in its present form, is too narrow and it doesn’t add much knowledge to the mechanism of action Metformin, if anything it adds confusion. As stated by the authors, macrophages can be beneficial and detrimental in any given disease depending on their activation status. Previous studies (mentioned in the introduction of the article) argue for a beneficial effect of Metformin during type 2 diabetes through the enhancement of the alternative activation of macrophages and/or suppression of their pro-inflammatory activities. Now the authors show that Metformin induces apoptosis in macrophages; but wouldn’t this be counterintuitive in terms of explaining the drug’s mode of action if it is killing the cells that mediate the therapeutic effects? Presumably, Metformin will be killing the M1s and enhancing (directly or indirectly) the M2 phenotype as I discussed in my previous review. In fact, this is what the authors hypothesise at the end of the introduction and yet fail to demonstrate: Thus, we hypothesized that metformin could attenuate inflammatory responses by inducing apoptosis of the inflammatory macrophages. This is precisely why apoptosis and activation of Metformin-treated macrophages should have been studied in parallel at the single cell level simply by flow cytometry for instance and not only in bulk analysis. This extra layer of understanding will broaden the article’s interest and impact. 

Response 1: Thanks for this comment and your concern. There are three possible apoptosis process in different phenotype macrophages after the metformin treatment: metformin kills the M1 only, M2 only, or both M1 and M2 phenotype macrophages. We fully agree with the reviewer that Metformin could kill the M1 phenotype macrophages. Killing M1 phenotype macrophages will reduce polarization of M1 phenotype macrophages to M2 phenotype macrophages. As shown in the Figure 7. A working model of the anti-inflammatory mechanism for metformin on macrophages, we have proposed a novel anti-inflammatory mechanism of metformin on macrophages by inducing apoptosis mainly in the M1 phenotype macrophages, presumably. We do not believe that metformin could enhance (directly or indirectly) the M2 phenotype macrophages.

Unfortunately, we cannot perform additional experiment at this time and we will consider this good suggestion in future research. However, we have added the following to the revised manuscript:

In the Discussion section:

Metformin might induce the apoptosis of M1 phenotype macrophages in this study. It has been demonstrated that metformin induced endoplasmic reticulum stress (Yang, Wei et al. 2015, Loubiere, Clavel et al. 2017), leading to apoptosis in M1, but not M2 macrophages. The transient Receptor Potential Canonical 3 (TRPC3) channel contributes to endoplasmic reticulum stress-induced apoptosis in M1, but not in M2 macrophages. TRPC3-deficient macrophages polarized to the M1 phenotype showed reduced apoptosis (Solanki, Dube et al. 2014, Solanki, Dube et al. 2017). In addition, the human cytolytic fusion proteins (CFP) also specifically eliminated polarized M1 macrophages in a transgenic mouse model of cutaneous chronic inflammation (Hristodorov, Mladenov et al. 2016). How metformin is related to TRPC3 and CFP needs further investigation.

References

Hristodorov, D., R. Mladenov, R. Fischer, S. Barth and T. Thepen (2016). "Fully human MAP-fusion protein selectively targets and eliminates proliferating CD64(+) M1 macrophages." Immunol Cell Biol 94(5): 470-478.

Loubiere, C., S. Clavel, J. Gilleron, R. Harisseh, J. Fauconnier, I. Ben-Sahra, L. Kaminski, K. Laurent, S. Herkenne, S. Lacas-Gervais, D. Ambrosetti, D. Alcor, S. Rocchi, M. Cormont, J. F. Michiels, B. Mari, N. M. Mazure, L. Scorrano, A. Lacampagne, A. Gharib, J. F. Tanti and F. Bost (2017). "The energy disruptor metformin targets mitochondrial integrity via modification of calcium flux in cancer cells." Sci Rep 7(1): 5040.

Solanki, S., P. R. Dube, L. Birnbaumer and G. Vazquez (2017). "Reduced Necrosis and Content of Apoptotic M1 Macrophages in Advanced Atherosclerotic Plaques of Mice With Macrophage-Specific Loss of Trpc3." Sci Rep 7: 42526.

Solanki, S., P. R. Dube, J. Y. Tano, L. Birnbaumer and G. Vazquez (2014). "Reduced endoplasmic reticulum stress-induced apoptosis and impaired unfolded protein response in TRPC3-deficient M1 macrophages." Am J Physiol Cell Physiol 307(6): C521-531.

Yang, J., J. Wei, Y. Wu, Z. Wang, Y. Guo, P. Lee and X. Li (2015). "Metformin induces ER stress-dependent apoptosis through miR-708-5p/NNAT pathway in prostate cancer." Oncogenesis 4: e158.

Reviewer 2 Report

Thank you for taking time and consideration of many comments. 

The major issue has not been solved. Addition of another macrophage model is required. Moreover, since the context of your study is cancer, it would be crucial to perform co-culture experiments which evaluate how metformin treated macrophages affect cancer cell properties. 

Author Response

Response to Reviewer 2 Comments

Point 1: Thank you for taking time and consideration of many comments.

The major issue has not been solved. Addition of another macrophage model is required. Moreover, since the context of your study is cancer, it would be crucial to perform co-culture experiments which evaluate how metformin treated macrophages affect cancer cell properties.

Response 1: Thanks for this comment. As we answered in the previous response, most of studies on the effects of metformin on macrophage only used one macrophage model (either human macrophages or mouse macrophages) and the chicken macrophages are also a good model. Metformin may be induced similar responses in both chicken and human macrophages according to the previous studies. Unfortunately, we cannot perform additional experiment at this time to confirm this speculation. However, we will consider the reviewer’s good suggestion in future research and have discussed this problem to the revised manuscript:

In the Discussion section:

Like humans or mice, chickens have been an important experimental system for developmental biology, immunology, and microbiology, leading to many fundamental discoveries (Brown, Hubbard et al. 2003). For example, chicken macrophages and human macrophages showed similar responses to Campylobacter jejuni, which signifcantly induced mRNA levels of inflammation-related signature genes in both chicken and human macrophages (Kim, Vela et al. 2018). The responses of chicken macrophages to CpG oligodeoxynucleotides (ODNs) also show similarities to human macrophages. ODN 2006, which has been reported to be an optimal stimulatory sequence for humans, also showed strong immunomodulatory effects on chicken macrophages, including increased secretion of the proinflammatory cytokine IL-6, enhanced NO2 release and upregulated cell surface marker expression (Xie, Raybourne et al. 2003). Whether metformin also induced similar responses in both chicken and human macrophages needs further investigation.

The purpose of this study was to investigate the effect of metformin treatment on macrophage apoptosis and explore the involvement of underlying pathways. The context of our study is not cancer. We are very sorry if whatever we wrote might have misled you to that.  So, it may not be necessary to perform these experiments in this study.

References

Brown, W. R., S. J. Hubbard, C. Tickle and S. A. Wilson (2003). "The chicken as a model for large-scale analysis of vertebrate gene function." Nat Rev Genet 4(2): 87-98.

Kim, S., A. Vela, S. M. Clohisey, S. Athanasiadou, P. Kaiser, M. P. Stevens and L. Vervelde (2018). "Host-specific differences in the response of cultured macrophages to Campylobacter jejuni capsule and O-methyl phosphoramidate mutants." Vet Res 49(1): 3.

Xie, H., R. B. Raybourne, U. S. Babu, H. S. Lillehoj and R. A. Heckert (2003). "CpG-induced immunomodulation and intracellular bacterial killing in a chicken macrophage cell line." Dev Comp Immunol 27(9): 823-834.
